# Multimodal Embodied Plan Prediction Augmented with Synthetic Embodied Dialogue

**Aishwarya Padmakumar**
Amazon
padmakua@amazon.com

**Mert İnan**[*]
Northeastern University
inan.m@northeastern.edu

**Spandana Gella**
Amazon
sgella@amazon.com

**Patrick L. Lange**
Amazon
patlange@amazon.com

**Dilek Hakkani-Tur**[*]
University of Illinois Urbana-Champaign
dilek@illinois.edu

## Abstract

Embodied task completion is a challenge where an agent in a simulated environment must predict environment actions to complete tasks based on natural language instructions and egocentric visual observations. We propose a variant of this problem where the agent predicts actions at a higher level of abstraction called a plan, which helps make agent actions more interpretable and can be obtained from the appropriate prompting of large language models. We show that multimodal transformer models can outperform language-only models for this problem but fall significantly short of oracle plans. Since collecting human-human dialogues for embodied environments is expensive and time-consuming, we propose a method to synthetically generate such dialogues, which we then use as training data for plan prediction. We demonstrate that multimodal transformer models can attain strong zero-shot performance from our synthetic data, outperforming language-only models trained on human-human data.

## 1 Introduction

Embodied task completion (Shridhar et al., 2020; Padmakumar et al., 2022) is a challenge where an agent in a simulated environment (Kolve et al., 2017; Savva et al., 2019; Chang et al., 2017) is given natural language context in the form of instructions or dialogue and needs to take actions in the environment to complete a desired task, additionally making use of egocentric visual observations. This typically requires the agent to predict actions directly executable in the simulated environment. For example, an action sequence to make coffee could start with actions to move forward two steps, turn left, and pick up a mug identified by a semantic segmentation mask. In contrast, physical robot systems tend to be more modular with a

---

[*] Contributions from Mert İnan and Dilek Hakkani-Tur were provided when they were employed at Amazon.

dedicated component for task planning - composing a sequence of fine-grained motor skills into a more complex task (Chen et al., 2010; Lemaignan et al., 2017; Jiang et al., 2019). In such a system, the coffee task considered above would likely start by invoking a semantic navigation module to find the mug and a grasping module to pick it up. Some prior work has been on embodied AI benchmarks suggesting that more modular models can outperform monolithic models (Min et al., 2021; Jia et al., 2022; Zheng et al., 2022; Min et al., 2022). However, these do not evaluate and explore the limitations of individual modules.

In this work, we formulate and explore the problem of task planning for embodied task completion. We improve upon existing plan prediction models and demonstrate the potential for improvement by comparing them with human plans. We adapt the Execution from Dialogue History (EDH) benchmark from the TEACh dataset (Padmakumar et al., 2022) to evaluate models at the level of a plan – a sequence of object interaction actions paired with the type of object on which the action needs to be executed – which are evaluated using task success upon execution with the aid of a heuristic plan execution module. Plan prediction is more challenging in TEACh compared to other embodied AI datasets, as tasks can be hierarchical and parameterized, environments are cluttered, and objects may be hidden inside closed receptacles. We evaluate variants of the multimodal Episodic Transformer (`E.T.`) model, previously used to directly predict low-level actions in embodied task completion (Shridhar et al., 2020; Padmakumar et al., 2022) and find that these outperform a finetuned language-only baseline.

Data collection for embodied AI tasks involving natural language is expensive and time-consuming to collect (Padmakumar et al., 2022), motivating the need for methods that require less human-human data. We develop the first framework for ex-

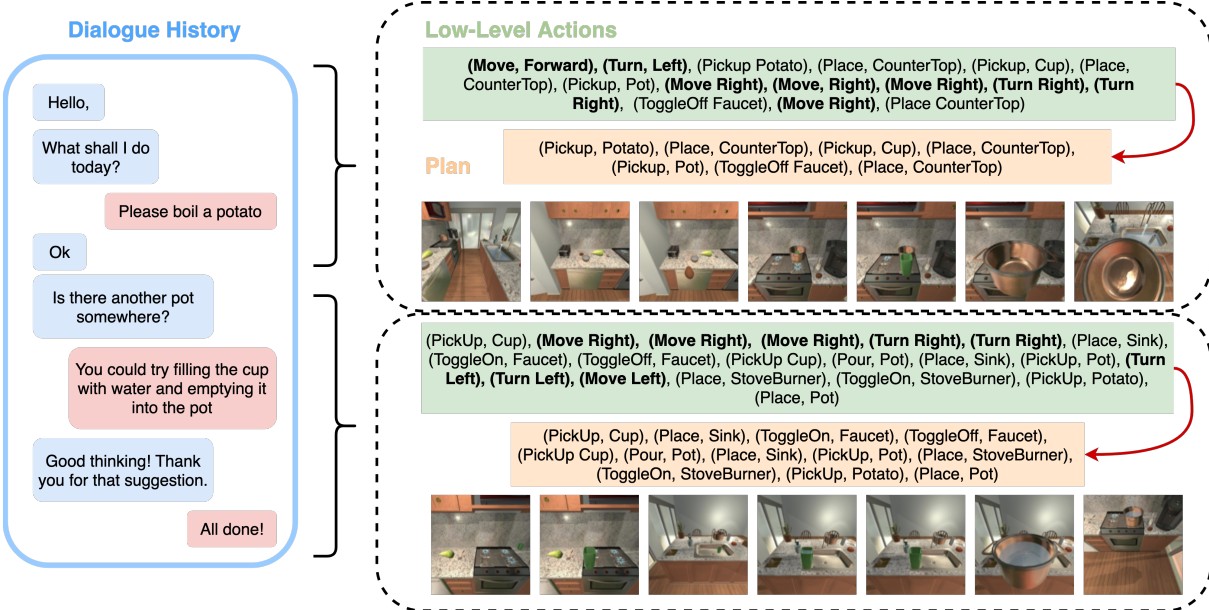

Figure 1: This figure depicts an example EDH instance from the TEACh dataset with modifications converting an action sequence to a plan. Models are trained to predict plans instead of low-level action sequences, and a plan execution module identifies navigation and other adjustment steps to ensure effective execution of plan actions.

panding agenda-based dialogue simulation (Schatzmann and Young, 2009) to a multimodal embodied agent setup by augmenting agenda-based dialogue act prediction with a rule-based module for identifying action sequences in the environment to complete tasks. We demonstrate that the E.T. based models trained only on synthetic data can achieve about 85% of the performance of the same models trained on real data, obtaining a zero-shot success rate of 17.09%, which outperforms the full shot success rate of plans generated by the language-only baseline at 10.27%.

To summarize, our contributions include:

- We formulate the problem of plan prediction for the TEACh dataset and evaluate a language-only baseline, variants of a multimodal transformer model (E.T.), and establish oracle performance on this problem.

- We are the first to design a framework for synthesizing embodied dialogues involving both utterances and environment actions to complete a task.

- We demonstrate that the synthetic data generated by our framework results in competitive zero-shot performance in our problem.

## 2 Task Setup

The TEACh dataset (Padmakumar et al., 2022) is an embodied dialogue dataset consisting of inter-

actions between human annotators role-playing a *Commander* and *Follower* collaborating in a simulated home environment to complete household tasks. Only the *Commander* has access to task information, and only the *Follower* can take actions in the environment requiring them to communicate to complete the task. An effective *Follower* must engage in dialogue with the *Commander*, obtain relevant information such as details of the task to be completed and locations of objects, and reason about environment actions that can accomplish relevant state changes to make progress in the task. We focus on the EDH benchmark from the TEACh dataset where given some dialogue history, as well as past actions and image observations, the *Follower* must predict subsequent actions in the environment to make progress with the task. This is evaluated by comparing environmental state changes arising from gold and predicted action sequences. We modify the expected prediction from a model to be a **plan**, which we define as a sequence of object interaction actions paired with the object category of the object they are to be executed upon [1]. An example of the task of boiling a potato is included in Figure 1. Note that in plan prediction, the model needs to reason about physical state changes - that the act of boiling requires

---

[1] While it is possible to specify more abstract plans, we choose this level of abstraction as the training data can be generated automatically from the TEACh EDH instances.

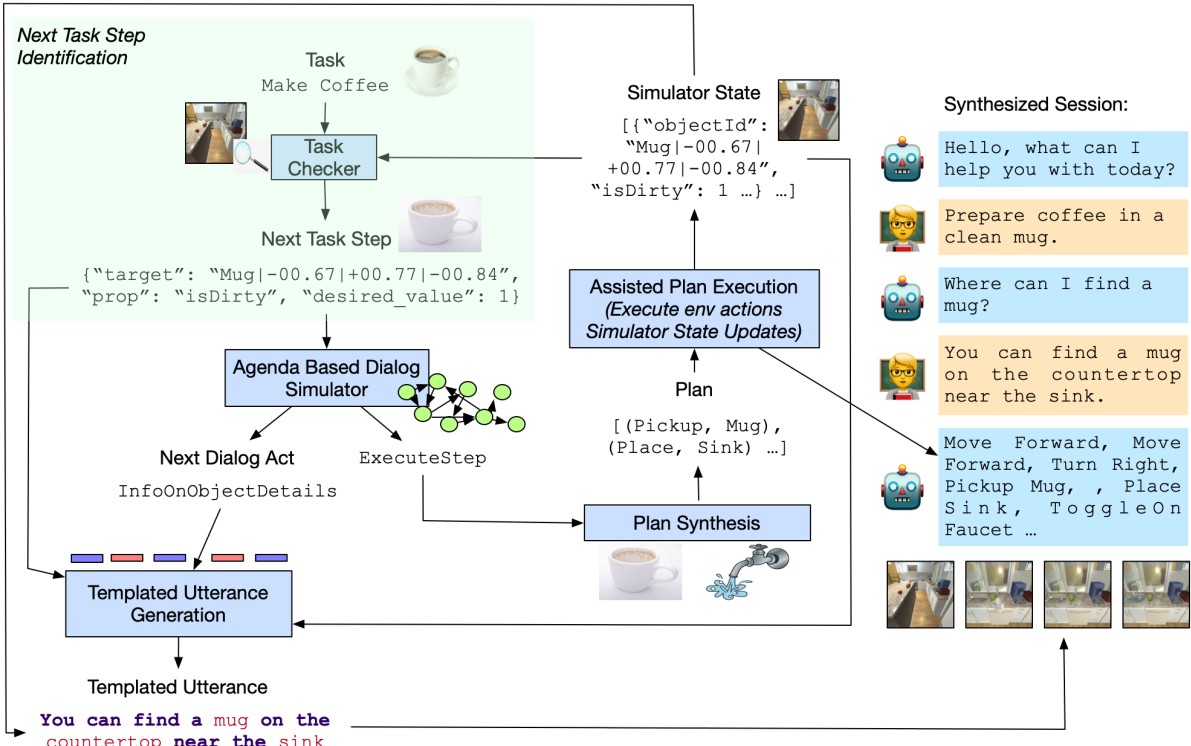

Figure 2: Control Flow for Simulating Synthetic Embodied Dialogues with an example of making coffee.

placing the potato in a container filled with water, which is then heated using a stove in the example. Other aspects of execution, such as navigating to required objects and fine-grained position adjustments, can be carried out by a separate execution module, which, in our case, is heuristic-based but can also be learned. This problem is non-trivial in datasets such as TEACh where tasks are parameterized, and hence highly variable, and diverse initial states can add or remove task steps. This is borne out in our experiments by the significant gap between model generated and human plans.

During inference, at each time step, a plan prediction model is expected to predict one object interaction action and the category of the object on which it is to be executed. This is then executed by one of two possible plan execution modules described in section 4. Execution terminates either when a model predicts a special `Stop` action, reaches a limit of 100 plan steps, or results in 30 simulator execution failures. A plan step may fail execution for a variety of reasons. It may be infeasible (e.g., trying to pick up a cabinet), a prerequisite step may not be completed (e.g., the `Slice` action is only feasible if the agent is holding a knife), or the agent may be poorly positioned (e.g., too close to the fridge to open it).

# 3 Plan Prediction Models

We adapt the Episodic Transformer (`E.T.`) model (Pashevich et al., 2021) for plan prediction. This is a multimodal transformer model which, in our case, receives the EDH dialogue history as language input and egocentric image observations as visual input and predicts plan steps consisting of an object interaction action and the type of object the action is to be taken on[2]. We obtain training data by filtering EDH action sequences to contain only object interaction actions and train the model as in Padmakumar et al. (2022), where the model receives images and plan steps from the EDH action history as input and predicts the entire output plan at once. At inference time, the last plan step predicted is executed, and the input for the next time step is updated with an image observation after executing this plan step. We use three variants of this model.

- **E.T.** : `E.T.` model as described above.
- **E.T. Hierarchical**: `E.T.` is modified to pass output from the action prediction head as input to the object prediction head.
- **E.T. + Mask**: Uses predefined constraints to determine whether the predicted action is

---

[2]See Appendix D for a more detailed explanation and a figure.

feasible to execute on the predicted object, and if not, backs up to the action with the highest probability that is feasible.

# 4 Plan Execution

While we can compare predicted plans with a ground truth plan using surface metrics such as edit distance, we believe a stronger test of predicted plans is executing them in the environment and measuring task success. To do this, we pair our models with heuristic plan execution modules:

- **Direct Plan Execution**: Given a predicted object interaction action and object type, we use object coordinates from the simulator to identify the closest object of the type [3], use the navigation graph to navigate to it and attempt to execute the action.

- **Assisted Plan Execution**: Direct plan execution can fail for various reasons. For example, if the sink is full and something needs to be placed in it, the sink needs to be cleared first. Although we train our plan prediction model with data that should enable it to predict such intermediate steps, we wish to explore whether models perform better if they abstract out such details. To do this, we analyze common execution failure cases and implement a set of heuristics, detailed in Appendix C, to increase their success.

In future work, we hope to replace these with models for plan executions (Zheng et al., 2022).

# 5 Synthetic Dialogue Generation

Collecting embodied dialogue examples is expensive and time-consuming (Padmakumar et al., 2022). However, it would be desirable for embodied agents to adapt to new tasks without requiring human interaction data. Ideally, given a task definition, we would like to be able to bootstrap a model for the task, which can then be further refined using techniques such as reinforcement learning. This work proposes a method to generate synthetic embodied dialogues to train an initial model without human interaction data. Our process for simulating synthetic embodied dialogues is outlined in Figure 2 and in the following sections. Additional details are included in Appendix G. We plan to release our synthetic data for future research.

## 5.1 Next Task Step Identification

Our dialogue simulation process involves breaking down a task into task steps corresponding to a single desired object state change, around which dialogue utterances and environment actions are constructed. Tasks in the TEACh dataset are defined as sets of object properties that must be satisfied for the task to be complete (Padmakumar et al., 2022). The public TEACh simulation wrapper [4] also includes a task checker that, when given a task definition and the current state of the simulator, can provide pending object state changes that need to be accomplished for the task to be considered complete. For example, Figure 2 demonstrates the synthesis of a dialogue session related to making coffee, which requires a mug in the environment to be clean and filled with coffee. As the agent acts in the environment, the task checker examines mugs in the environment and identifies the one closest to completion. The task checker can indicate to the agent the object state changes that still need to be accomplished on this mug; for example, in Figure 2, it identifies a face that needs to be cleaned. This is a single `Task Step` that can be used as a focus for dialogue utterances and environment actions. Once this `Task Step` is completed, the simulation process proceeds to the next `Task Step` of filling coffee, after which the task checker will indicate task completion, ending the dialogue simulation.

## 5.2 Agenda Based Dialogue Simulation

Given the next `Task Step`, we build a semantic outline for a snippet of synthetic dialogue related to this `Task Step` that includes dialogue acts exchanging information related to this `Task Step` and predicting a special action `ExecuteStep` that indicates a transition to predicting environment actions that accomplish this `Task Step` [5].

We do this by building an **agenda-based dialogue simulator** (Schatzmann and Young, 2009) over dialogue acts relevant to the TEACh dataset combined with the `ExecuteStep` action. We use a subset of the dialogue acts annotated for the TEACh dataset in Gella et al. (2022), focusing on requesting and receiving instructions related to the task and how to complete it, as well as the locations of objects. Our agenda-based simulator consists of 9 dialogue states, each computed as

---

[3]Note that using the closest object is a heuristic and can fail. In our experiments, we evaluate two oracle plan prediction methods to quantify the limitations of this.

[4]https://github.com/alexa/teach

[5]Note that human-human dialogue in TEACh is much more free-form and we hope to achieve more versatility in future work.

a boolean function of 8 binary dialogue features. We pre-define probabilities for sampling dialogue acts, `ExecuteStep`, and `DoNothing` actions in each state. We then generate a dialogue session by sampling a sequence of actions, expanding each dialogue act into an utterance using templates filled in with task and simulator information, and each `ExecuteStep` into a sequence of environment actions as described in section 5.3. Appendix G includes more details of this process.

## 5.3 Plan Synthesis

When we predict that the session should transition from dialogue to environment actions, we use a rule-based system to identify an action sequence in the environment that is likely to accomplish the next `Task Step`. We hard code plans for each possible object state change, detailed in Appendix G. For example, if the object state change requires an object to be cleaned, the plan will involve moving the object to the sink, turning on and off the tap, picking it up, and pouring out water accumulated from cleaning. These hard-coded plans do not account for handling difficulties arising from the current state of the environment, for example, clearing out the sink if it is too full to place the object to be cleaned. Hence, we execute these synthesized plans using Assisted Execution (Section 4) to improve our success rate at completing task steps using these hard-coded plans.

## 6 Experiments

### 6.1 Plan Prediction Models

We evaluate our proposed plan prediction models on the EDH task of the TEACh dataset (Padmakumar et al., 2022). We experiment with each of the models in section 3 with each execution method in section 4. Additionally, we evaluate the following baseline and oracle conditions [6]:

**Baseline:** Our baseline is a language-only BART model (Lewis et al., 2020), finetuned to take in the EDH dialogue history and predicts the entire plan as a sequence of tokens that are post-processed for validity and executed as in (Gella et al., 2022).

---

[6]We do not compare to TEACh EDH models in this paper (Padmakumar et al., 2022) as our execution methods access information that the TEACh baseline models cannot access. We do this to ensure that we are evaluating only the process of plan prediction without additional complications arising from navigation and simulator behavior

**Oracle:** As an upper bound to the success rate obtainable with each of our plan execution methods, we obtain oracle plans using the ground truth actions in the EDH instance. We filter these action sequences, retaining only object interaction steps and converting object IDs to object types to match the plan representation used by our models.

**Oracle with Object IDs (`CorefOracle`):** To further test the limitations of our plan representation combined with the heuristic of selecting the closest object of a particular type, we produce plans from human action sequences containing object IDs instead of object types to avoid ambiguity during plan execution.

We additionally include our best zero-shot model and our best model trained on both real and synthetic data. These models use synthetic data generated according to the method outlined in Section 5 using the initial states corresponding to the TEACh train set. The zero-shot model is trained only on synthetic data, and the data-augmented model is trained on a combination of real and synthetic data.

We evaluate models based on the success rate (SR) and goal condition success rate (GC) as defined in the original TEACh paper (Padmakumar et al., 2022). Success rate, which measures the fraction of EDH instances for which predicted plans produced all expected object state changes, and GC, which measures the fraction of such object state changes across all instances, were calculated. Since the TEACh test set is not public, we follow the standard protocol proposed in the TEACh codebase [7] of using a standardized division of the original validation sets into validation and test sets called the divided validation and divided test sets, each of which is further divided into *Seen* and *Unseen* splits.

We present our results in Table 1. For a subset of these conditions, we train and perform inference with three random seeds and perform 2-sided Welch t-tests. Allowing for Bonferroni corrections over four tests, we find that `E.T. + Mask` is trending to be significantly better than the baseline with $p = 0.0381$ on the `divided_val_seen` split and $p = 0.0164$ on the `divided_test_seen` split. We did not find any statistically significant difference between the `E.T. Hierarchical` and `E.T. + Mask` models [8].

---

[7]https://github.com/alexa/teach

[8]We did not perform statistical comparisons across all pairs of conditions as it is expensive and time-consuming to run an

| Model | Execution | EDH Plan Divided Val Split | | | | EDH Plan Divided Test Split | | | |
|---|---|---|---|---|---|---|---|---|---|
| | | *Seen* | | *Unseen* | | *Seen* | | *Unseen* | |
| | | SR | GC | SR | GC | SR | GC | SR | GC |
| Baseline | Direct | 11.26 | 13.67 | 7.51 | 11.03 | 7.19 | 9.62 | 8.87 | 9.54 |
| | Assisted | 11.92 | 17.27 | 8.91 | 12.19 | 9.80 | 12.30 | 10.27 | 12.31 |
| E.T. | Direct | 12.91 | 16.32 | 15.58 | 16.20 | 15.03 | 19.52 | 16.62 | 15.61 |
| | Assisted | 15.89 | 20.57 | 18.74 | 22.36 | 16.67 | 19.96 | 19.98 | 27.13 |
| E.T. Hierarchical | Direct | 14.24 | 15.67 | 16.23 | 17.27 | 14.71 | 17.97 | 17.27 | 20.30 |
| | Assisted | 18.21 | 20.45 | 18.09 | 24.53 | 17.97 | 23.67 | 19.70 | 25.82 |
| E.T. + Mask | Direct | 15.23 | 22.51 | 17.81 | 18.29 | 16.34 | 23.84 | 17.46 | 18.96 |
| | Assisted | **18.87** | 28.99 | **19.57** | 27.64 | **18.95** | 26.35 | 20.07 | 28.33 |
| Best zero shot | Assisted | **18.87** | 17.52 | 16.23 | 19.30 | 15.36 | 17.17 | 17.09 | 16.82 |
| Best Augmented | Assisted | **18.87** | 26.90 | 19.48 | 30.41 | 17.32 | 26.52 | **22.32** | 34.30 |
| Oracle | Direct | 61.92 | 63.64 | 55.57 | 58.48 | 54.58 | 53.58 | 56.77 | 58.01 |
| | Assisted | 68.87 | 72.13 | 61.97 | 63.07 | 61.44 | 62.49 | 63.21 | 64.87 |
| CorefOracle | Direct | 77.81 | 83.03 | 70.87 | 71.87 | 75.82 | 79.50 | 71.90 | 74.34 |
| | Assisted | 80.13 | 84.50 | 74.58 | 77.31 | 78.43 | 80.92 | 76.94 | 78.30 |

Table 1: Success rate (SR) and Goal Condition Success Rate (GC) of different models combined with different execution methods on the TEACh EDH task. Oracle performances are separated as upper bounds on the task. Best performance results are bolded for each metric and split in the specific execution method.

Since oracle plans do not obtain a 100% success rate, we observe the limitations of our plan execution method, which can only handle 78.43% of unseen test instances even with object coreference resolved (`CorefOracle`). We believe this is due to the difficulty in obtaining perfect positioning and placement even with ground truth simulator information, further supported by the gap between direct and assisted execution of oracle plans. We additionally see that there is considerable scope for improvement from resolving ambiguity related to which object instance to manipulate, which accounts for an improvement of about 17% between `Oracle` and `CorefOracle`.

We observe that while the `E.T.`, `E.T. Hierarchical` and `E.T. + Mask` models substantially improve over the baseline, there is also a considerable gap between them and `Oracle` which uses the same plan representation, which demonstrates that there is considerable scope for improvement in understanding the details of the task to be completed from dialogue, and reasoning about actions to take to achieve the corresponding state changes. Qualitatively, we find that multimodal input's main benefits are breaking down complex

tasks such as watering a plant, for which detailed steps are rarely provided in the dialogue, and identifying how much of the task has already been completed. Failures of the `E.T.` models mainly arise from not learning when to stop, which is a limitation of the current inference procedure. Other causes of failure include performing unrelated manipulations on easily visible objects or ignoring small objects in favour of larger ones.

On comparing models trained on real data with synthetic data, we find that the zero-shot models perform surprisingly well, outperforming the baseline trained on real data and approaching the performance of the models that have the same architecture but are trained on real data. This suggests that when generalizing to new tasks for this application, it might be reasonable to train a model purely on synthetic data and expect reasonable performance from interaction with human users.

### 6.2 Zero Shot Training Ablation

We perform further ablations to identify how zero-shot model performance varies according to data size in Table 2. While we see a trend towards improvement in performance with increasing data size, and the best results are obtained at higher data sizes, the improvement is not perfectly con-

inference with enough random seeds to allow for Bonferroni corrections as the number of tests grows.

| Model | Train Set Size | EDH Plan Divided Val Split | | | | EDH Plan Divided Test Split | | | |
| | | Seen | | Unseen | | Seen | | Unseen | |
| | | SR | GC | SR | GC | SR | GC | SR | GC |
|---|---|---|---|---|---|---|---|---|---|
| E.T. | 1x | 12.25 | 9.50 | 13.17 | 9.33 | 14.05 | 11.04 | 13.82 | 9.36 |
| E.T. | 2x | 15.89 | 13.95 | 14.47 | 13.58 | **14.38** | 12.03 | 15.97 | 11.85 |
| E.T. | 3x | **17.88** | 11.94 | **14.84** | 10.61 | 13.07 | 12.14 | **16.06** | 11.22 |
| E.T. | 4x | 16.89 | 11.28 | 13.45 | 13.07 | 14.05 | 11.26 | 14.94 | 13.54 |
| E.T. Hierarchical | 1x | 15.56 | 10.15 | 13.36 | 10.77 | 11.76 | 9.73 | 12.32 | 9.74 |
| E.T. Hierarchical | 2x | 15.89 | 11.16 | 14.19 | 16.68 | 11.44 | 13.50 | 14.47 | 18.53 |
| E.T. Hierarchical | 3x | **18.87** | 17.52 | **16.23** | 19.30 | **15.36** | 17.17 | **17.09** | 16.82 |
| E.T. Hierarchical | 4x | 18.21 | 10.45 | 14.75 | 10.14 | 14.05 | 11.97 | 15.69 | 11.21 |
| E.T. + Mask | 1x | 13.58 | 9.98 | 13.82 | 10.90 | 13.73 | 12.30 | 13.35 | 9.83 |
| E.T. + Mask | 2x | 14.24 | 16.39 | **15.49** | 16.08 | **14.38** | 12.41 | **15.59** | 14.94 |
| E.T. + Mask | 3x | **16.89** | 12.05 | 14.10 | 12.45 | 12.09 | 11.21 | 15.03 | 12.09 |
| E.T. + Mask | 4x | 15.23 | 14.37 | 14.66 | 15.20 | **14.38** | 13.45 | 14.29 | 14.98 |

Table 2: We explore how zero shot model performance varies with the size of the synthetic training set (as a proportion of the size of the human-human training data). This table reports success rate (SR) and goal condition success rate (GC) on the EDH divided test split with assisted execution.

| Model | Synthetic Data Size | EDH Plan Divided Val Split | | | | EDH Plan Divided Test Split | | | |
| | | Seen | | Unseen | | Seen | | Unseen | |
| | | SR | GC | SR | GC | SR | GC | SR | GC |
|---|---|---|---|---|---|---|---|---|---|
| E.T. | 1x | **18.21** | 17.52 | 17.90 | 29.12 | 17.65 | 23.67 | **21.38** | 31.63 |
| E.T. | 2x | 17.55 | 25.00 | **19.02** | 24.99 | **18.63** | 23.51 | 19.70 | 24.60 |
| E.T. | 4x | 17.22 | 26.84 | 18.37 | 26.12 | 17.97 | 25.26 | 19.79 | 26.01 |
| E.T. Hierarchical | 1x | 16.89 | 24.70 | **18.74** | 25.73 | 18.30 | 23.46 | **20.45** | 25.65 |
| E.T. Hierarchical | 2x | **18.54** | 19.48 | **18.74** | 23.55 | **20.92** | 28.05 | 19.61 | 27.91 |
| E.T. Hierarchical | 4x | 14.90 | 20.31 | 17.90 | 24.22 | 17.97 | 26.95 | 20.17 | 24.30 |
| E.T. + Mask | 1x | **18.87** | 26.90 | **19.48** | 30.41 | 17.32 | 26.52 | **22.32** | 34.30 |
| E.T. + Mask | 2x | 16.89 | 32.84 | 19.02 | 30.66 | 18.63 | 27.99 | 20.45 | 29.98 |
| E.T. + Mask | 4x | 18.21 | 27.38 | 18.65 | 28.01 | **20.26** | 27.88 | 19.79 | 27.28 |

Table 3: We explore how synthetic data size impacts model performance. Models here are trained on a combination of the human-human TEACh training data and synthetic data of the size indicated in column Synthetic Data Size (as a proportion of the size of the human-human training data). This table reports success rate (SR) and goal condition success rate (GC) on the EDH divided test split with assisted execution.

sistent with data size. We believe this sub-linear scaling with the increase in data is due to a combination of limitations in the range of TEACh initial states in which our synthetic plan execution method can generate a valid action sequence successfully, limitations of the diversity in synthetic templates and limitations in the ability of the E.T. model to model the TEACh plan prediction task effectively. We believe that through engineering improvements or a learned Reinforcement Learning policy, we can

improve the range of initial states covered; with the assistance of better LLMs, we can produce more diverse synthetic dialogue, and using neural SLAM models, we can overcome the limitations of what a particular model can learn. We plan to explore these directions in future work.

Note that we did not perform other hyperparameter tuning for models in Table 2 besides the changes in training data. The best condition, `E.T. Hierarchical` with a synthetic training set of size 3x as large as the human-human training set from Table 2 has been reported in Table 1 for comparison with models trained on human-human data. We find that this model trained purely on synthetic data obtains a success rate of 17.09% on the divided unseen test split, outperforming the language-only baseline trained on real data at 8.87%, and approaches close to the performance of `E.T. Hierarchical` trained on human-human data at 19.70%.

## 6.3 Data Augmentation Training Ablation

In Table 3, we ablate different sizes of synthetic data when used in data augmentation. We find that when both real and synthetic data are included, larger sizes of the synthetic training set are less beneficial than when trained only on synthetic data. The best condition `E.T. + Mask` with a combination of human-human and synthetic data of equal size has been included in Table 1 as "Best Augmented." We find that at 22.32% on the divided unseen test split, this slightly outperforms the same model condition trained on human-human data at 20.07%, and is much stronger than the best condition using only synthetic data at 17.09%.

## 7 Related Work

**Task Planning:** Interactive systems on physical robots typically have a modular structure in which task planning plays a significant role (Chen et al., 2010; Khandelwal et al., 2017; Peshkin et al., 2001). In simulated environments, Logeswaran et al. (2022) propose a language-only finetuned GPT-2 model for task planning on ALFRED . Some end-to-end ALFRED models also have task planning as a component (Min et al., 2021; Jia et al., 2022; Blukis et al., 2022). However, this is a simpler dataset where task planning can be cast as a 7-way classification problem. Prior work has also explored language-only task planning using finetuned BART models in TEACh (Gella et al., 2022;

Zheng et al., 2022; Zhang et al., 2022), which we compare to as a baseline.

**Dialogue Simulation:** User simulation in the dialogue community originally consisted of rule-based systems designed using linguistic knowledge to enable finetuning dialogue systems to individual user preferences and subsequently evolved into trainable probabilistic models that can be used to bootstrap a dialogue system in the initial development phase and further finetune it through reinforcement learning (Schatzmann et al., 2006; Young et al., 2013). A common method for building user simulators is agenda-based simulation (Schatzmann and Young, 2009), which uses a predefined set of transition probabilities between dialogue acts in combination with goal information to sample subsequent dialogue acts. This has been used to bootstrap a range of dialogue models ranging from probabilistic POMDP models (Schatzmann et al., 2007) and text-to-SQL models (Liu et al., 2022) to hierarchical deep reinforcement learning methods (Peng et al., 2017). In this work, we augment a standard agenda-based simulator with an additional intent to determine transitions to acting in the environment to generate situated dialogues. Another popular paradigm for dialogue simulation is to develop two models - one for the user and one for the agent side and train them simultaneously using reinforcement learning (Liu and Lane, 2017; Shah et al., 2018). This has also been used for some multimodal dialogue domains (Das et al., 2017), but we choose not to adapt it in our domain as the time to generate a single dialogue is higher in situated applications due to the latency from executing environment actions in the simulator.

## 8 Conclusions and Future Work

We develop a model for multi-modal plan prediction for the TEACh dataset using the Episodic Transformer architecture and evaluate end-to-end performance on the TEACh EDH task in conjunction with heuristic plan execution modules. We additionally experiment with training this model using only synthetic data generated using an agenda-based dialogue simulator combined with an environment action generator that uses a combination of rules and simulator information to identify sequences of actions in the environment that can make progress with the task. We find that our `E.T.` plan prediction models outperform a BART baseline, even when BART is finetuned on human-

human embodied dialogue data but `E.T.` is fine-tuned only on synthetic data, suggesting that our dialogue simulation approach is a viable alternative to expensive data collection for bootstrapping an embodied task completion model on new tasks. We also find a considerable performance gap between models and humans in plan prediction that cannot be easily closed by techniques such as data augmentation.

The recent success of large language models in a wide variety of structured prediction tasks, including robotic planning tasks similar to ours, is a promising future direction for our work. Some preliminary attempts prompting large language models indicate that it is non-trivial to design an appropriate representation of the dialogue history and plans for LLMs to guess at the remaining objects required for subsequent actions effectively. Additionally, while it is likely that recent long context LLMs can likely take in a significant amount of state information of the environment, further work is required to determine the best way to provide this in environments such as AI2-THOR, especially if a proposed method much scale up to more realistic home environments with larger numbers of objects. It is likely also possible to improve the diversity of the generated simulated data by paraphrasing generated utterances using LLMs.

## 9   Limitations

In this paper, we explore the problem of plan prediction for embodied task completion, and conduct our experiments in the TEACh  dataset, which is set in the AI2-THOR simulator. While we hypothesize that models developed for plan prediction will transfer better to physical robots as they align better with levels of abstraction at which robotics systems are currently implemented, further experimentation is needed to evaluate such transferability. In the short term, such experiments will likely need to work on problems with a simpler action space as some of the *Follower* actions supported in AI2-THOR, such as slicing, are not supported in most robots available currently. Additionally, it would be beneficial to test plan prediction models and our dialogue simulation method on similar tasks set in other simulators. This is a direction we plan to pursue in future work. We believe this is beyond the scope of this publication due to the considerable engineering effort involved in adapting models across different embodied AI simulators and task

representations.

In this work, we describe a method to generate synthetic dialogues for embodied task completion by augmenting an agenda-based dialogue simulator with modules that break up an embodied AI task into individual object state changes, and rule-based methods to identify actions that complete them. Additionally, we currently manually define transition probabilities between dialogue acts for simulation, which would also need to be modified for a new dataset. While our approach is effective at bootstrapping plan prediction models without any human-human interaction data, it requires simulator-specific engineering, particularly when defining heuristics for assisted plan execution. Additionally, for both plan prediction inference and dialogue simulation, every action must be executed in the simulator. This results in considerable compute time, as discussed in the appendix. While dialogue simulation does not require the use of a GPU, the plan prediction models do - both for training and inference. We have additionally found that the Episodic Transformer models used in the paper show a noticeable variance in performance when trained with the same hyperparameters but with different random seeds. We attempt to strengthen our conclusions by training models with multiple seeds for statistical analysis where it is beneficial, but we would also like to highlight the development of models with less variance as an important direction for future research.

Finally, our work is currently limited to English as we are not familiar with datasets in other languages that provide language instructions for tasks that require complex reasoning over multi-step action sequences.

## 10   Ethics Statement

This work is part of a broader tradition of building natural language interfaces to control various devices. Natural language interfaces such as language-based search and intelligent personal assistants provide convenience and have the potential to make multiple forms of technology ranging from mobile phones and computers, as well as robots or other machines such as ATMs or self-checkout counters more accessible and less intimidating to users who are unfamiliar or uncomfortable with other interfaces on such devices such as command shells, button-based interfaces or changing visual user interfaces. Spoken language interfaces can

also be used to make such devices more accessible for the visually impaired or users who have difficulty with fine motor control.

User trust in such interfaces is essential. Depending on the circumstances, therefore, some considerations to keep in mind are: (1) whether the collection of personal data benefits the user; (2) whether the collection of personal data is transparent to the user; (3) whether the user understands whether and how they can control the collection of personal data; and (4) whether the user has the ability to elect whether to use such interfaces, opt into or out of the collection of certain personal data, access and update certain personal data, and delete certain personal data. Additionally, safeguards may be required as embodied agents become capable of interacting with arbitrary objects in the world to reduce the likelihood of accidents or malicious misuse.

In this work, we also experiment using simulated embodied dialogue data to train models. On one hand, the use of simulated data can limit the collection of personal data. On the other hand, simulators may not be designed to represent the full range of user behavior and may perform better for some users than others.

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

## A  Licensing and Responsible Use

In this section, we discuss the licensing and usage of existing scientific artifacts in this paper:

- TEACh dataset: The TEACh dataset is released under a CDLA-Sharing V 1.0 license, with images released under Apache 2.0 and code under an MIT license. We believe our usage does not violate any of these license terms. We do not redistribute any of these as part of our work as they are publicly available. The TEACh dataset was created to study models for translating natural language instructions combined with egocentric visual observations into action sequences. Our use case is a subtask of this intended use case. We also believe using the TEACh code to generate synthetic dialogue sessions is consistent with this goal.

- AI2-THOR simulator: The AI2-THOR simulator (Kolve et al., 2017) is necessary to use the TEACh dataset and generate synthetic dialogue sessions. We believe our usage of AI2-THOR does not violate the license. We plan to release our data under the CDLA-Sharing V 1.0 license, with images released under Apache 2.0, which we believe will be consistent with AI2-THOR and TEACh license

terms. We believe our work is consistent with the intended usage of AI2-THOR, which is to further embodied AI research in the household domain.

- BART model: We finetune BART (Lewis et al., 2020) as a baseline for our task, which is released under the Apache 2.0 license. We believe our usage is consistent with the license and do not intend to redistribute it. We believe that our use is consistent with the intended usage of BART as a general purpose sequence to sequence language model.

- Episodic Transformer Model: We use the Episodic Transformer model (Pashevich et al., 2021), which is released under an MIT License, for plan prediction with some architectural modifications. We believe our usage is consistent with the intent of this model, which was designed for a very similar embodied AI application in a similar dataset.

The TEACh dataset was manually inspected by the original authors to remove identifying information and offensive utterances (Padmakumar et al., 2022). Since our generated data is templated, we do not believe this can contain personally identifying information or offensive utterances. Our synthetic dialogues are set in the TEACh environment and only cover the tasks listed in the TEACh dataset. The dialogues are in English and are mainly intended to cover requesting and informing of task steps (in the form of object state changes) and locations of objects. They support a limited breakage of strict turn-taking by allowing an agent to not generate an utterance at some time steps in the generation procedure.

## B  Additional Related Work

**Task Planning on Physical Robots**  Task planning has long been a standard component of physical robot architectures (Chen et al., 2010), particularly with general purpose service robots (Khandelwal et al., 2017; Peshkin et al., 2001). Classical task planners include a symbolic representation of the state of the world, a goal, and skills the robot can execute. They are expected to find a sequence of skills that, when executed, will transform the world into the goal state, typically using heuristic search algorithms (Lipovetzky, 2014). Over the years, research in planning has improved the symbolic

representations used in planners (Fikes and Nilsson, 1971; McDermott, 1996; Gelfond and Kahl, 2014; Konidaris et al., 2018; Gopalan et al., 2020), search algorithms (Hart et al., 1968; Helmert, 2004; Richter and Westphal, 2010) and handling uncertainty via probabilistic methods (Toussaint and Goerick, 2007; Bagchi et al., 1996; Ponzoni Carvalho Chanel et al., 2019). More recent work has focused on expanding beyond fully defined world representations by expanding to use common sense (Al-Moadhen et al., 2013) and open worlds (Jiang et al., 2019). Some interesting efforts in this direction use Large Language Models to perform planning (Ahn et al., 2022).

**Embodied AI Tasks in Simulation**  Simulated environments (Kolve et al., 2017; Savva et al., 2019; Puig et al., 2018; Chang et al., 2017; Yan et al., 2018) have been used over recent years to explore the efficacy of deep learning methods that directly use egocentric visual observations instead of data from expensive sensors. While there is a challenge in transferring results from simulated to real environments, simulated environments are more accessible, less expensive, and allow for the testing of technologies that may not be sufficiently safe for use in the real world (Savva et al., 2019). Additionally, while simulated environments can be used for tasks that do not require the use of language (Anderson et al., 2018a; Batra et al., 2020; Gan et al., 2020; Kant et al., 2022), they play a particularly valuable role in developing language understanding and reasoning capabilities over actions that are currently difficult for physical robots to complete, but we hope it will become a reality in the future (Kolve et al., 2017). Much of the work in language understanding for embodied AI happens using vision and language navigation, where an agent must learn to navigate through a previously unseen environment purely based on natural language route instructions (Anderson et al., 2018b; Chen et al., 2019; Thomason et al., 2020). Embodied task completion additionally requires performing object manipulation actions (Shridhar et al., 2020; Padmakumar et al., 2022; Suhr et al., 2019; Kim et al., 2020; Narayan-Chen et al., 2019). In this work, we specifically focus on the TEACh dataset (Padmakumar et al., 2022) as it involves more complex tasks that require non-trivial task planning.

## C  Assisted Plan Execution

This section outlines the full set of heuristics involved in assisted plan execution:

- For all actions, if the target object property change is already complete, do nothing to avoid an execution failure.

- *Pickup:* If the object is inside a receptacle (container), open the receptacle. After pickup, if a receptacle was opened, close it.

- *Place:* If the target receptacle is in a receptacle, take it out and place it on the counter first (for example, if we need to place something on a plate that is inside a drawer). If the target receptacle needs to be opened, open it and close it after placement (for example, a drawer or microwave needs to be opened to place something inside). If a placement attempt fails, try removing the existing contents of the receptacle one by one to make more space.

- *Open, Close:* Toggle off the target object if relevant (for example, microwaves need to be turned off to open them).

- *ToggleOn, ToggleOff:* If the target is open, close it first (for example, microwaves need to be closed to turn them on).

- *Slice:* If the target is in a receptacle, move it to the counter first.

Additionally, we also attempt position adjustments to increase the chance of success.

## D  `E.T.` model

This section discusses the Episodic Transformer (`E.T.`) Model (Pashevich et al., 2021) along with our modifications in greater detail. For convenience, we include a diagram of the model in Figure 3. The `E.T.` model receives language (in our case, EDH dialogue history) and egocentric image observations of size 900 x 900 as input. Visual observations are first resized to 224 x 244 and then encoded using a visual encoder that is based on a Faster R-CNN model trained on 325K frames of expert demonstrations from the ALFRED train fold (which comprises seen splits of TEACh ) and not finetuned in any of our experiments. This encoder average-pools ResNet features four times and adds

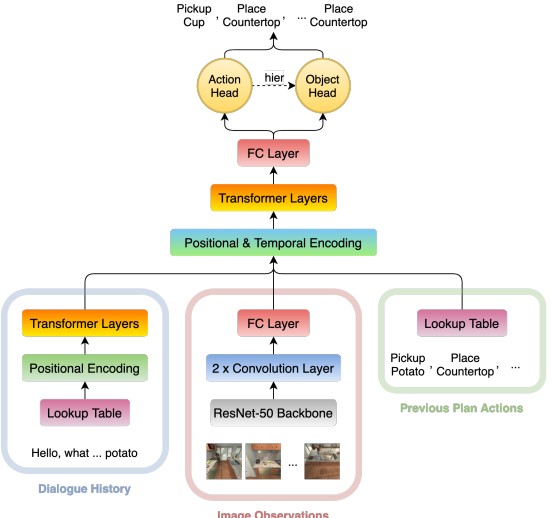

Figure 3: This depicts the architecture for the `E.T.`-based models. The basic E.T. model does not have a connection between the action and object heads, but `E.T. Hierarchical` does. There are three main input processing components (dialogue history, image observation, and previous plan actions). These are then input into the positional and temporal encodings and transformer layers to develop the next feasible set of high-level plan actions.

a dropout of 0.3 to obtain feature maps of 512 x 7 x 7. These are then fed into two convolutional layers with 256 and 64 filters of size 1 x 1, respectively, and mapped using a fully connected layer to size 768.

The language input is tokenized using `revtok` [9] and encoded using two transformer layers with 12 attention heads and an embedding size of 768, which are trained from scratch. The language and visual encodings are concatenated and passed through two multimodal transformer layers with 12 attention heads, which are also trained from scratch in our experiments.

To predict actions and objects, the final embedding corresponding to each image from the input is projected using fully connected layers and then passed to two independent softmax prediction heads over the action and object space, respectively. At inference time, the last action, which is not a padding token, is identified and used for prediction, along with the object, at the same time step. The `E.T. Hierarchical` model connects the two prediction heads by passing the output of the action head as an additional input to the object head. The `E.T. + Mask` model examines the action-object

---

[9] https://pypi.org/project/revtok/

pair predicted at inference time, and if they are found to be incompatible, replaces the action with the action of highest probability with the predicted object. The object is assumed to be the more reliable prediction as it is more likely to be directly visible.

## E  Plan Prediction Model Hyperparameters

For training the `E.T.`, `E.T. + Mask` and `E.T. Hierarchical` methods, we retained hyperparameters from the original TEACh paper (Padmakumar et al., 2022), except the batch size, without further hyperparameter tuning, and used the largest batch size that could fit in a single GPU of a p3.8xlarge AWS EC2 instance. Note that the following hyperparameters were kept constant for all experiments reported in this paper, and results in different tables arise only from changes in training data, model choice (between `E.T.`, `E.T. Hierarchical` and `E.T. + Mask`), and plan execution method. We used the AdamW optimizer with 0.33 weight decay with a learning rate of 1e−4 for the first 10 epochs and 1e−5 for the last 10 epochs. We trained all models for 20 epochs with a batch size of 3 and reported results from the final epoch. We replace sampling with rotation permutations of our training dataset per epoch, ensuring that every training example is seen exactly once in our dataset. For the language decoder in the transformer, we use a dropout of 0.1, and for the encoder, we use a dropout of 0.1. The different `E.T.` models required 4 hours for preprocessing (extracting image features using the ResNet-50 backbone) and about 2 hours per model for training using 4 GPUs of a p3.8xlarge AWS EC2 instance. At inference time, we could use a maximum of 3 GPUs for inference as one GPU was required by the simulator. When using 3 GPUs of a p3.8xlarge AWS EC2 instance, E.T. models took about 11 hours to complete inference jointly on the `divided_val_seen` and `divided_test_seen` splits and about 35 hours to complete inference jointly on the `divided_val_unseen` and `divided_test_unseen` splits. The time difference is due to the size of the various splits.

For the baseline BART model, we retain hyperparameters from the model presented in (Gella et al., 2022). We take the pretrained BART-base model from the Huggingface library [10] and finetune for 20 epochs using a batch size of 2 per

[10]https://huggingface.co/

GPU. The training was done using gradient accumulation across 4 GPUs of a p3.8xlarge AWS EC2 instance. We use the AdamW optimizer with $\beta_1 = 0.9$, $\beta_2 = 0.99$, $\epsilon = 1e08$ and weight decay of 0.01. We use a learning rate of 5e05 with a linear warmup over 500 steps. The BART model can be finetuned in under an hour using all 4 GPUs of a p3.8xlarge AWS EC2 instance. We first performed inference on the BART model and saved the predicted plans to file before separately executing them in the AI2-THOR simulator. This process can also be completed in under an hour. Executing stored plans either in the case of the BART model or the oracle conditions took about 2.5 hours using 3 GPUs of a p3.8xlarge AWS EC2 instance for the combined `divided_val_seen` and `divided_test_seen` splits and about 8 hours for the combined `divided_val_unseen` and `divided_test_unseen` splits.

## F  Data Statistics

The number of games and EDH instances in the TEACh data splits and batches of synthetic data used in this paper are included in Table 4.

| Split | Number of games | Number of EDH instances |
|---|---|---|
| TEACh train | 3121 | 3895 |
| TEACh divided_val_seen | 83 | 302 |
| TEACh divided_val_unseen | 309 | 1078 |
| TEACh divided_test_seen | 98 | 306 |
| TEACh divided_test_unseen | 303 | 1071 |
| Synthetic 1x | 1587 | 2034 |
| Synthetic 2x | 3272 | 3875 |
| Synthetic 3x | 4952 | 7201 |
| Synthetic 4x | 6360 | 9284 |

Table 4: Data statistics in TEACh and synthetic data splits.

## G  Dialogue Simulation Details

This section provides a more detailed description of the dialogue simulation process. Dialogue simulation begins by sampling which agent starts the interaction - the *Commander* or the *Follower*, each

with 50% While it is possible to create new initial states, we iterate over each initial state in the train split of the TEACh dataset, each of which is associated with a task to be completed in that state. For each agent, the *Commander* and the *Follower* we maintain a state for the agent that is factored into binary state features. We then use predefined probabilities for sampling different dialogue acts in each state and alternate taking turns between the two agents. In this implementation, we use the following dialogue acts defined in Gella et al. (2022):

- RequestForInstruction

- Instruction

- RequestForObjectLocationAndOtherDetails

- InfoOnObjectLocationAndOtherDetails

- Acknowledge

- FeedbackPositive

- FeedbackNegative

However, we use both FeedbackPositive and FeedbackNegative only to end the dialogue either as a success or failure respectively. We additionally divide the Instruction dialogue act into two sub-types for convenience:

- Instruction: For communicating the task and its parameters

- Step: For communicating a single desired object state change that would result in progress towards completion of the task, for example, cleaning a mug to fill it with coffee eventually.

We correspondingly also create a special RequestStep action. Besides this, we have two non-dialogue actions that an agent can also choose to perform:

- ExecuteStep: This is a cue to transition to actions in the environment. This is only performed by the *Follower* which identifies a rule-based plan to accomplish the desired state change and executes it with assisted plan execution described in section 4.

- DoNothing: This allows an agent to skip a turn and hence avoids rigid turn-taking in the resultant dialogue and introduces variability in the amount of information communicated in the dialogue to mimic real dialogues better.

The state features used are:

- dialogue_started: Agents start with this feature set to False, indicating that the agent is in the initial state before any dialogue has taken place and must initiate dialogue. It gets set to True once an initial utterance has been exchanged.

- goal_communicated: This feature is False initially and set to True after an Instruction utterance has been sent from the *Commander* to the *Follower* communicating the high level task.

- cur_step_requested: This feature is False initially, gets set to True when the *Follower* sends a RequestStep action to the *Commander* and reset to False when environment actions are executed.

- cur_step_sent: This feature is False initially, gets set to True when the *Commander* sends the current step to the *Follower* through a Step dialogue act and reset to False when environment actions are executed.

- cur_step_obj_requested: This feature is False initially, gets set to True when the *Follower* requests the location of an object and reset to False when environment actions are executed.

- cur_step_obj_sent: This feature is False initially, gets set to True when the *Commander* sends the location of an object and reset to False when environment actions are executed.

- task_complete: This feature is False initially and gets set to True when the task is completed successfully.

- follower_stuck: This feature is used to identify failed dialogues. It is False initially and set to true if the *Follower* attempts environment actions but is unable to accomplish the intended object state change. When this is identified, the dialogue is terminated early.

Given the value of the state features, the next state of the agent is computed as boolean functions over state features included in Table 5. Given the dialogue acts we then sample dialogue acts according to Table 6 for the *Commander* and Table 7 for the *Follower*.

| dialogue_not_started | ¬ dialogue_started |
|---|---|
| goal_or_step_start | dialogue_started ∧¬ goal_communicated ∧¬ task_complete ∧¬ follower_stuck ∧¬ cur_step_requested ∧¬ cur_step_sent ∧¬ cur_step_obj_requested ∧¬ cur_step_obj_sent |
| step_start | dialogue_started ∧ goal_communicated ∧¬ task_complete ∧¬ follower_stuck ∧¬ cur_step_requested ∧¬ cur_step_sent ∧¬ cur_step_obj_requested ∧¬ cur_step_obj_sent |
| step_requested | dialogue_started ∧¬ task_complete ∧¬ follower_stuck ∧ cur_step_requested ∧¬ cur_step_sent |
| step_sent | dialogue_started ∧¬ task_complete ∧¬ follower_stuck ∧ cur_step_sent ∧¬ cur_step_obj_requested ∧¬ cur_step_obj_sent |
| obj_loc_requested | dialogue_started ∧¬ task_complete ∧¬ follower_stuck ∧ cur_step_obj_requested ∧¬ cur_step_obj_sent |
| obj_loc_sent | dialogue_started ∧¬ task_complete ∧¬ follower_stuck ∧ cur_step_obj_sent |
| task_complete | dialogue_started ∧ task_complete ∧¬ follower_stuck |
| follower_stuck | dialogue_started ∧¬ task_complete ∧ follower_stuck |

Table 5: Boolean functions over dialogue state features to compute current dialogue state (¬ represents NOT and ∧ represents AND).

| | | |
|---|---|---|
| dialogue_not_started | Instruction | 0.8 |
| | Step | 0.1 |
| | DoNothing | 0.1 |
| goal_or_step_start | Instruction | 0.8 |
| | Step | 0.1 |
| | DoNothing | 0.1 |
| step_requested | Step | 0.9 |
| | DoNothing | 0.1 |
| step_start | Step | 0.9 |
| | DoNothing | 0.1 |
| obj_loc_requested | InfoOnObject Details | 0.9 |
| | DoNothing | 0.1 |
| follower_stuck | Feedback Negative | 1.0 |
| task_complete | Feedback Positive | 1.0 |

Table 6: Probabilities for sampling various dialogue acts for the *Commander* given the dialogue state.

| | | |
|---|---|---|
| dialogue_not_started | RequestFor Instruction | 1.0 |
| goal_or_step_start | RequestStep | 0.8 |
| | ReqForObjLoc AndOD | 0.1 |
| | ExecuteStep | 0.1 |
| step_start | RequestStep | 0.8 |
| | ReqForObjLoc AndOD | 0.1 |
| | ExecuteStep | 0.1 |
| step_requested | ExecuteStep | 1.0 |
| step_sent | ReqForObjLoc AndOD | 0.9 |
| | ExecuteStep | 0.1 |
| obj_loc_requested | ExecuteStep | 1.0 |
| obj_loc_sent | ExecuteStep | 1.0 |

Table 7: Probabilities for sampling various dialogue acts for the *Follower* given the dialogue state.

Given a dialogue act, we use the following templates to get utterances:

- RequestForInstruction: Hello, what can I help you with today?

- Instruction: This is filled in with the description field from the task definition, for example Make coffee or Put all Forks in any Sink.

- RequestStep: Done. What should I do next?

- Step: This is filled it from the condition_failure_desc field from the TEACh task definition. For example, The Mug does not have coffee. or The Fork must be placed in a Sink.

- RequestForObjectLocationAndOtherDetails: Where can I find a/an ⟨object⟩? where ⟨object⟩ is identified from the task step.

- InfoOnObjectDetails: You can find a/and ⟨object⟩ in/on a/an ⟨object⟩ near a/an ⟨object⟩ where the reference objects are identified using location information in the simulator.

- FeedbackPositive: Great! We're all done.

If an ExecuteStep action is sampled, we then synthesize the plan for the property to be changed as part of the current task step using Table 8. This is then executed using assisted execution (Section 4, Appendix C) to obtain the final sequence of environment actions. If this produces the desired object state change, dialogue simulation continues; otherwise, it is terminated by entering the follower_stuck state. Our final implementation of dialogue simulation successfully simulates a dialogue that completes the task in 35.4% of the initial states in the TEACh dataset.

| Desired Outcome | Plan |
|---|---|
| Cleaning | (Pickup, ⟨TARGET⟩), (Place, Sink), (ToggleOn, Faucet), (ToggleOff, Faucet), (Pickup, ⟨TARGET⟩), (Pour, Sink) |
| Making Coffee | (Pickup, ⟨TARGET⟩), (Place, CoffeeMachine), (ToggleOn, CoffeeMachine) |
| Slicing | (Pickup, Knife), (Slice, ⟨TARGET⟩), (Place, CounterTop) |
| Toasting Bread | (Pickup, ⟨TARGET⟩), (Place, Toaster), (ToggleOn, Toaster), (Pickup, ⟨TARGET⟩), (Place, CounterTop) |
| Cooking | (Pickup, ⟨TARGET⟩), (ToggleOff, Microwave), (Open, Microwave), (Place, Microwave), (Close, Microwave), (ToggleOn, Microwave), (ToggleOff, Microwave), (Open, Microwave), (Pickup, ⟨TARGET⟩), (Close, Microwave), (Place, CounterTop) |
| Placing | (Pickup, ⟨TARGET⟩), (Place, ⟨DESIRED_VALUE⟩) |
| Boiling | (Pickup, ⟨TARGET⟩), (Place, Bowl), (Pickup, Bowl), (Place, Sink), (ToggleOn, Faucet), (ToggleOff, Faucet), (Pickup, Bowl), (ToggleOff, Microwave), (Open, Microwave), (Place, Microwave), (Close, Microwave), (ToggleOn, Microwave), (ToggleOff, Microwave), (Open, Microwave), (Pickup, ⟨TARGET⟩), (Close, Microwave), (Place, CounterTop) |
| Fill With Water | (Pickup, Cup), (Place, Sink), (ToggleOn, Faucet), (ToggleOff, Faucet), (Pickup, Cup), (Pour, ⟨TARGET⟩) |
| Turn On | (ToggleOn, ⟨TARGET⟩) |
| Open | (Open, ⟨TARGET⟩) |

Table 8: Plans to accomplish object state changes