# OpenReview forum: "Multimodal Embodied Plan Prediction Augmented with Synthetic Embodied Dialogue"
_EMNLP/2023/Conference — EMNLP 2023 Main_

### Official Review · Reviewer_MiCJ · 2023-08-01

**Soundness:** 3

**Excitement:**

3: Ambivalent: It has merits (e.g., it reports state-of-the-art results, the idea is nice), but there are key weaknesses (e.g., it describes incremental work), and it can significantly benefit from another round of revision. However, I won't object to accepting it if my co-reviewers champion it.

**Paper Topic And Main Contributions:**

The paper addresses the problem of embodied task completion, where an artificial agent in a simulated environment predicts actions to complete tasks based on natural language instructions and egocentric visual observations. The authors propose a variant of this problem where the agent makes predictions at a higher level of abstraction, which they refer to as a "plan". The paper presents results showing that multimodal transformer models trained on synthetic data can achieve strong zero-shot performance.

**Questions For The Authors:**

In addition to the above comments in the reason to reject section, would it be possible to generate similar synthetic data by prompting LLM? Comparing with the synthetic dialogue generated by the proposed method, what would be the advantages of the proposed method in generating synthetic dialogue?

**Reasons To Accept:**

The paper formulates the problem of plan prediction for the TEACh dataset, and generates synthetic dialogues for the task. The experiment shows that the model trained on the synthetic data can achieve good zero-shot performance. The way of generating synthetic dialogues could be generalized to help solving other tasks.

**Reasons To Reject:**

The proposed plan prediction task is not novel where it is already adopted in prior works on TEACh dataset, eg Film: Following instructions in language with modular methods and Jarvis: A neuro-symbolic commonsense reasoning framework for conversational embodied agents, where they predict subgoals before generate low-level actions.

Also, the evaluation misses large language models where they are likely to achieve good performance on the plan prediction task with simple in-context learning.

**Reproducibility:**

4: Could mostly reproduce the results, but there may be some variation because of sample variance or minor variations in their interpretation of the protocol or method.

**Reviewer Confidence:**

4: Quite sure. I tried to check the important points carefully. It's unlikely, though conceivable, that I missed something that should affect my ratings.

---

> ### Author Rebuttal · Authors · 2023-08-29
>
> We thank the reviewer for their time and insights. We would like to highlight that we do mention plan prediction as being an important subtask for existing methods such as JARVIS and FILM. However, these works do not explicitly evaluate this subproblem or quantify its difficulty for the TEACh or ALFRED datasets. In ALFRED, plan prediction can be reduced to a simple 7-way classification, but in TEACh, this is a significantly more challenging problem due to the increased complexity of the tasks and the range of task parameters (which we found to be very difficult to predict accurately from the TEACh dialogs).
> Additionally, we had explored generating the dialogue entirely, as well as paraphrasing synthetic dialogue using OPT models, but we found that this often led to inaccuracies in details of the task such as the exact object to be used, resulting in worse downstream performance. We can include these negative results in our appendix if accepted and can also repeat the experiments with more recent open source LLMs.
> We do think that this would be a good testbed for prompting LLMs, although it is a good question to decide how much ground truth information about the environment setup should be made visible to the LLM. Some preliminary experiments we performed in this direction with OPT models did not lead to any successful plans but we hope to iterate on this prompting approach in future work.

---

### Official Review · Reviewer_zTkK · 2023-08-03

**Soundness:** 3

**Excitement:**

3: Ambivalent: It has merits (e.g., it reports state-of-the-art results, the idea is nice), but there are key weaknesses (e.g., it describes incremental work), and it can significantly benefit from another round of revision. However, I won't object to accepting it if my co-reviewers champion it.

**Paper Topic And Main Contributions:**

This paper deals with prediction of high level plan (sequence of actions) for a given dialogue history, past actions and image observations. They present a synthetic dialogue generation method, which they use to then train multimodal transformers.
They show that multimodal models trained on this synthetically generated data outperforms text-based baseline models that are trained on real data.

**Reasons To Accept:**

This paper addresses the issue of low availability of training data for a complex multimodal task.
The author show how properly generated synthetically training data can help improve the performance of multimodal transformer over model trained on real data.

**Reasons To Reject:**

1. Paper was a relatively hard reading for me and I feel would not be easily accessible to readers who do not experience in this field.
2. The task variant is a simplification of the actual TEACh task, as such performance of the proposed method is not yet proven for the actual task that better reflects the actual challenges of the task.

**Reproducibility:**

3: Could reproduce the results with some difficulty. The settings of parameters are underspecified or subjectively determined; the training/evaluation data are not widely available.

**Reviewer Confidence:**

1: Not my area, or paper was hard for me to understand. My evaluation is just an educated guess.

---

> ### Author Rebuttal · Authors · 2023-08-29
>
> We thank the reviewer for their time and insights. We apologize that our paper was difficult to a reader less experienced in this field and would appreciate constructive suggestions on how to revise the text to make it more broadly accessible if the paper is accepted.
> Additionally, while we show improvements for only the TEACh EDH plan prediction task, we do show in Table 1 that this in itself is a fairly challenging task with the TEACh baseline having about 20% test unseen success rate but human plans having about 63% success rate.
> We additionally believe that our synthetic data generation method is a valuable resource beyond its use in TEACh tasks, particularly in the exploration of other embodied AI problems for which no manually collected action sequence data exist.

---

### Official Review · Reviewer_YKo6 · 2023-08-05

**Soundness:** 4

**Excitement:**

4: Strong: This paper deepens the understanding of some phenomenon or lowers the barriers to an existing research direction.

**Missing References:**

Modular solution on ALFRED:
- A Persistent Spatial Semantic Representation for High-level Natural Language Instruction Execution (https://arxiv.org/abs/2107.05612)

Modular solution on TEACh:
- DANLI: Deliberative Agent for Following Natural Language Instructions (https://arxiv.org/abs/2210.12485)

**Paper Topic And Main Contributions:**

This paper studies the prospect of using machine-generated embodied dialog and trajectory to replace/augment human-generated dialog/trajectory for embodied task learning. Specifically, the authors use the ground-truth task progress information provided by the simulator (TEACh in AI2THOR), paired with a template-based embodied dialog generation pipeline, and a rule-based action planner to obtain purely machine-generated dialog exchanges and action trajectories. The authors evaluate the effectiveness of this generated data on a simplified version of the TEACh EDH task (where navigation is removed from the task, the agent only needs to predict object manipulation actions and object class). The model used for comparison of human-generated data (original TEACh) and the machine-generated data is an end-to-end model (Episodic Transformer, E.T.) that takes in visual observation, dialog history, and previous actions and predicts the next plan step. Encouragingly, the author finds that models trained with machine-generated data only perform on par with models trained with human-generated data, and mixing machine- and human-generated data together could outperform using human-only data. This finding suggests the effectiveness of using machine-generated data for embodied dialog.

**Questions For The Authors:**

Q1: The paper would benefit from a bit more discussion on why the "data scaling law" analyses in Table 2 and 3 don't follow a strictly increasing pattern. Could it be stemmed from the lack of diversity in the dialog template used to generate the data?

Q2: I suggest the authors refer to the EDH used in this paper as "EDH-Plan-Only" since the EDH has been simplified in this paper (see L197-198) by giving the agent the ground-truth coordinate of the object thus removing the need to learn navigation. Using the term EDH might mislead some readers to draw a direct comparison between the results in this paper and other papers that uses the full EDH setting. Note that this is a suggestion and it does not negatively influence my rating of this paper.

**Reasons To Accept:**

S1: Important topic. Data scarcity has been one of the major bottlenecks in embodied AI research. The reliance on humans to collect such dialog data, especially in an embodied setting, is expensive and not scalable. This paper sheds some initial light on how machine-generated data could help alleviate this problem.

S2: Solid engineering work. The author described the process of generating the dialog and trajectory data concisely in the paper, but having worked in this domain myself, I know the tremendous amount of engineering work that must have gone into this. Good job!

S3: Thorough experiments, convincing results. It is encouraging to see that this type of human-generated data shows some initial signs of life. The "data scaling law" analysis in Table 2 and 3 is also helpful, although I think it could use a bit more analysis and discussion (see Q1 below).

**Reasons To Reject:**

No strong reason to reject.

**Reproducibility:**

4: Could mostly reproduce the results, but there may be some variation because of sample variance or minor variations in their interpretation of the protocol or method.

**Reviewer Confidence:**

5: Positive that my evaluation is correct. I read the paper very carefully and I am very familiar with related work.

---

> ### Author Rebuttal · Authors · 2023-08-29
>
> We would like to thank the reviewer for their time and insights, particularly the acknowledgement of the engineering effort that goes into works such as this.
>
> To answer the reviewer’s questions:
>  - Q1: We believe the sub-linear scaling with the increase in data is due to a combination of limitations in the range of TEACh initial states in which our synthetic plan execution method is able to successfully generate a valid action sequence, limitations of the diversity in synthetic templates and limitations in the ability of the ET model to effectively model the TEACh plan prediction task. We believe that through engineering improvements, or a learned Reinforcement Learning policy, we can improve the range of initial states covered, with the assistance of better LLMs we can produce more diverse synthetic dialogue and using neural SLAM models, we can overcome the limitations of what a particular model can learn. We plan to explore these directions in future work and hope that the reviewer will consider the current results sufficient for an initial publication.
>  - Q2: We will make the suggested change to EDH-Plan_Only to avoid confusion with the original TEACh EDH task.
>
> We will also include the additional suggested references in our related work.

---

### Meta-Review · Area_Chair_Qgcs · 2023-09-19

**Recommendation:** 5

**Metareview:**

The paper studies the use of machine-generated dialog-trajectory pairs for training symbolic plan prediction models in embodied trajectory-from-dialog tasks. The paper proposes a method to generate this data from the simulator, and finds that a plan-prediction transformer model trained on this data is on par with human-generated data.

Some strengths cited by individual reviewers:
“The paper solves an important problem of data scarcity in embodied AI research”.
“Solid engineering work.”
The synthetic data generation pipeline is general with some modification
“Thorough experiments and convincing results”

Outstanding concerns after rebuttal:
- Only looked at training transformers, did not try LLM prompting. My opinion: although LLM prompting is a very popular approach and deserves being looked at, it is also its own moving target that this paper did not aim to study.
- Section describing data-generation was hard to read for reviewer zTkK, who is not in the area. Constructive feedback was discussed

Overall, there is consensus that the paper is sound and has addressed the research question it sought to study. Reviewer YKo6 gave a soundness score of 4 with the highest confidence, while the other reviewers scored it as 3.

Excitement scores are 4 from reviewer YKo6 (the most confident and well-matched reviewer), and 3 from everyone else.

---

### Decision · Program_Chairs · 2023-10-07

**Decision:**

Accept-Main

**Comment:**

The paper studies the use of machine-generated dialog-trajectory pairs for training symbolic plan prediction models in embodied trajectory-from-dialog tasks. The paper proposes a method to generate this data from the simulator, and finds that a plan-prediction transformer model trained on this data is on par with human-generated data.

Some strengths cited by individual reviewers:
“The paper solves an important problem of data scarcity in embodied AI research”.
“Solid engineering work.”
The synthetic data generation pipeline is general with some modification
“Thorough experiments and convincing results”

Outstanding concerns after rebuttal:
- Only looked at training transformers, did not try LLM prompting. My opinion: although LLM prompting is a very popular approach and deserves being looked at, it is also its own moving target that this paper did not aim to study.
- Section describing data-generation was hard to read for reviewer zTkK, who is not in the area. Constructive feedback was discussed

Overall, there is consensus that the paper is sound and has addressed the research question it sought to study. Reviewer YKo6 gave a soundness score of 4 with the highest confidence, while the other reviewers scored it as 3.

Excitement scores are 4 from reviewer YKo6 (the most confident and well-matched reviewer), and 3 from everyone else.